# Multi-channel coupling of decay instability in three-dimensional low-beta plasma

Horia Comişel[1,2], Yasuhito Narita[3,4], and Uwe Motschmann[1,5]

[1]Institut für Theoretische Physik, Technische Universität Braunschweig, Mendelssohnstr. 3, D-38106 Braunschweig, Germany
[2]Institute for Space Sciences, Atomiştilor 409, P.O. Box MG-23, Bucharest-Măgurele, RO-077125, Romania
[3]Space Research Institute, Austrian Academy of Sciences, Schmiedlstr. 6, A-8042 Graz, Austria
[4]Institut für Geophysik und extraterrestrische Physik, Technische Universität Braunschweig, Mendelssohnstr. 3, D-38106 Braunschweig, Germany
[5]Deutsches Zentrum für Luft- und Raumfahrt, Institut für Planetenforschung, Rutherfordstr. 2, D-12489 Berlin, Germany

**Correspondence:** H. Comişel
(h.comisel@tu-braunschweig.de)

**Abstract.** Three-dimensional hybrid simulations have been carried out to verify the hypothesis of simultaneous multi-channel decay of a large-amplitude Alfvén wave in a low-beta plasma, e.g., in the shock-upstream region or the solar corona. Obliquely-propagating daughter modes are excited along the perpendicular direction to the mean magnetic field at the same parallel wavenumbers and frequencies with the daughter modes driven by the field-aligned decay. We find that the transversal spectrum of waves is controlled by the multi-channel coupling of the decay process in low-beta plasmas and originates in the dispersion state of the shear Alfvén wave.

**Keywords.** Space plasma physics (Wave-wave interactions, Wave-particle interactions, Numerical simulation studies).

## 1 Introduction

Quasi-parallel propagating Alfvén waves with circular polarization are frequently observed in various space plasma domains, in particular in the vicinity of the Earth's bow shock or other planetary bow shocks (see e.g., Narita et al., 2007), where the shock-reflected ions transfer energy from the ion beam into electromagnetic waves through the ion beam instability (often right-hand resonance in which the gyration of the beam ions is observed as right-hand polarized in the rest frame of the core or thermal plasma). The beam-excited waves attain large amplitudes up to the saturation level of the beam instability and develop into a stage of wave-wave interactions (see e.g., Gary, 1991; Akimoto et al., 1993; Gomberoff et al., 2000; Wang and Lin, 2003; Li et al., 2013). Parametric instability is thought to play an important role in distributing the fluctuation energy of large-amplitude Alfvén waves into different modes and different wavelengths (see e.g., Terasawa et al., 1986; Hoshino and Goldstein, 1989; Spangler et al., 1997; Nariyuki and Hada, 2006; Bekhor and Drake, 2003).

The decay instability is a type of parametric instability and is expected in both low- and high- beta plasmas (see e.g., Inhester, 1990; Vasquez, 1995). We point out that the decay instability in three-dimensional nature has multiple channels at once and can excite daughter waves both in the Alfvén mode and in the sound (or ion acoustic) mode without further cascade of the daughter waves or before the cascade occurs and the wave develops into turbulence. The reason for the multi-channel coupling lies on the fact that the decay instability in 3-D low-beta plasmas occurs simultaneously in various direction imposed by the mean magnetic field, and has a degeneration over the directions perpendicular to the mean field. In the analytic dispersion analysis, one typically sets a priori the propagation direction of the daughter wave and studies the growth rate of the decay instability in the wavenumber domain (see e.g., Derby, 1978; Goldstein, 1978; Longtin and Sonnerup, 1986; Wong and Goldstein, 1986; Hollweg, 1994; Ruderman and Simpson, 2004; Araneda et al., 2007; Brodin and Stenflo, 2015). The situation is different in a higher-dimensional set-up, that is, the daughter waves can be excited in various directions at once in spirit of Boltzmann's equal-probability over all the possible channels. The degeneration of the decay instability over the perpendicular directions is illustrated in Fig. 1. Wave-wave coupling in the decay instability forms a parallelogram in the frequency-wavenumber domain (parallel to the mean magnetic field) such that the both energy quantum $\hbar\omega$ and the momentum quantum $\hbar k$ are conserved during the wave decay. The pump Alfvén wave (propagating parallel to the mean field, denoted by $A_1$, the first-order magnetic field after the zero-th order mean field) decays into a backward or anti-parallel propagating Alfvén mode ($A_2$, the second-order field) and a forward propagating sound wave $S_2$. Wave decay in the parallel direction has a degeneracy over the perpendicular directions. The degeneracy can be resolved by plotting parallelogram-type wave couplings in the wavevector domain spanning the parallel component $k_\parallel$ and the perpendicular component $k_\perp$. That is, the momentum quantum is conserved in the vectorial sense, $\hbar\boldsymbol{k} = \mathrm{const}$. One may write the individual realization of the wave decay as '0' for the exactly parallel decay, '+' and '-' for a slightly oblique decay, '++' and '- -' for an even more oblique decay, and so on.

Earlier studies on the nonlinear interaction between Alfvén waves and obliquely-propagating waves (see e.g., Mjølhus and Hada, 1990; Viñas and Goldstein, 1991a, b; Laveder et al., 2002; Nariyuki et al., 2008) report that the growth rates of the decay instability in the oblique direction to the mean magnetic field are typically smaller than the case of field-aligned parametric instabilities. Numerical simulations such as multi-dimensional MHD (magnetohydrodynamics) and hybrid plasma simulations lead to the conclusion with respect to the circular polarization that the parametric decay of large-amplitude Alfvén waves develops particularly along the propagation direction of the pump wave (see e.g., Del Zanna et al., 2001; Verscharen et al., 2012; Gao et al., 2013).

The nature of Alfvén wave decay into oblique propagation angles has extensively been studied. Viñas and Goldstein (1991ab), for example, argue in their analytical dispersion analysis with circularly polarized Alfvén waves that the oblique-propagation decay instability can compete against the field-aligned decay in the limit of small propagation or decay angles to the mean magnetic field in the low-beta plasma. Ghosh et al. (1993) confirm in their MHD 2-D simulations that obliquely-propagating daughter waves are excited by the decay of a field-aligned Alfvén wave for the low beta regime. The results by Ghosh et al. (1993) are based on a $\beta$-value of 0.5, but nevertheless they demonstrate that the field-aligned decay instability is persistent and dominates the oblique-decay instabilities. Matteini et al. (2010a) discovered by 2-D hybrid simulations that lin-early polarized Alfvén waves with oblique-direction of propagation decay in low beta plasmas in a broad spectrum of coupled

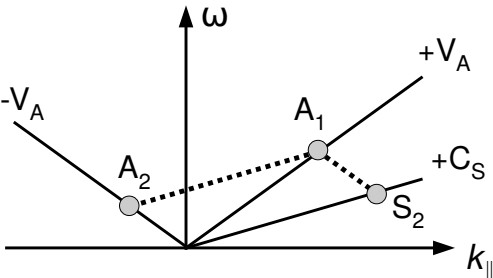

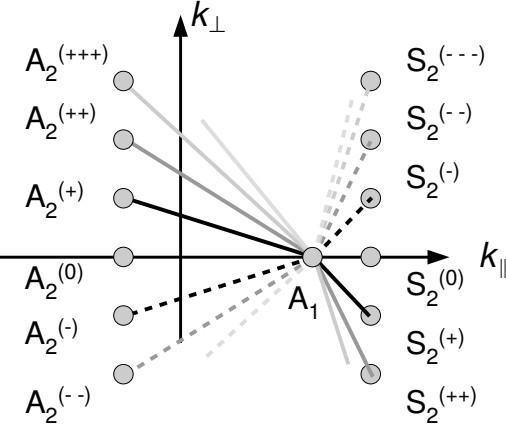

**Figure 1.** Three-wave couplings of the decay instability (parallelogram formation) in the frequency-wavenumber domain parallel to the mean magnetic field (top panel) and in the wavevector domain spanning parallel and perpendicular directions (bottom panel).

Alfvén waves and density fluctuations perpendicular to the direction of the mean magnetic field. Matteini et al. (2010a) also noticed that the magnetic daughter modes follow the dispersion relation for a shear Alfvén wave, $\omega = k_\| V_A$, where $\omega$, $k_\|$, and $V_A$ are the frequency, parallel wavenumber and Alfvén velocity, respectively.

Here we present a scenario that decay instability can occur simultaneously at various angles to the mean magnetic field, generating a number of second-order fluctuations or waves (after the pump wave being as the first-order fluctuation). We refer to the simultaneous decay as "multi-channel couplings" following the notion in the scattering theory, for example, coupled-channels in nuclear reactions, (see e.g., Tamura, 1969; Tobocman, 1975). Our goal is to study the hypothesis or the scenario of the multi-channel coupling by running a three-dimensional hybrid plasma simulation in a low-beta plasma setup.

## 2 Hybrid plasma simulation

### 2.1 Simulation run

A hybrid plasma simulation is carried out to verify the hypothesis of the multi-channel coupling in the low-beta decay instability. We use the hybrid code AIKEF (Müller et al., 2011) in a three-dimensional spatial configuration. The size of the simulation box on each direction is L=288 $d_i$, and $(576)^3$ computational cells are set, containing 1000 super-particles in each cell. The length scale is normalized to the ion inertia length $d_i = V_A/\Omega_p$, where $V_A$ and $\Omega_p$ are the Alfvén velocity and the ion gyro-frequency (for protons), respectively. The value of ion and electron betas used in the simulation are, $\beta_i = \beta_e = 0.01$. This low value of beta (the equivalent fluid beta in the simulation is $\beta$=0.02) assures a faster linear growth and an earlier non-linear saturation of the parametric instabilities. **The magnetic-field amplitude of the Alfvén pump wave (normalized to the value of the background magnetic field) has a value of 0.2.** The simulation is halted at a time of $t\Omega_p$=600 shortly after the saturation of the decay instability.

Figure 2 displays the time evolution of the fluctuation energy for the field-aligned pump wave (Alfvén mode $A_1$) which represents the Fourier mode $(m_\parallel, m_\perp) = (10, 0)$, $(m_{\parallel(\perp)} = k_{\parallel(\perp)}L/2\pi)$, and the daughters $A_2^{(0)}, A_2^{(-)}, A_2^{(--)}, A_3^{(---)}$, and, $S_2^{(0)}, S_2^{(-)}, S_2^{(--)}, S_3^{(---)}$, respectively, in the parallel and perpendicular wavenumber domain. Counter-propagating Alfvén daughter modes (to the pump wave propagation direction) and sound daughter modes start to develop simultaneously, and the fluctuation energy increases both along the mean magnetic field and in the oblique directions to the mean field. The mode number for the parallel-propagating Alfvén lower sideband mode is $m_\parallel = -9$ while the corresponding sound mode has a mode number of $m_\parallel = 19$, satisfying the three-wave coupling rule. An exponential growth of the fluctuation energy represents the linear stage of the decay instability. Both parallel (or field-aligned) propagating waves and obliquely-propagating waves grow simultaneously. The slope of the wave growth is nearly the same between the magnetic mode and the sound mode at lower propagation angles to the mean magnetic field (typically up to 40 degree). The decay instability becomes then saturated at a time of $t\Omega_p$=300. It is interesting to note that the pump Alfvén wave still dominates the fluctuation energy in the system. **The low level of the amplitude of the magnetic field and density daughter modes in Fig. 2 could be a consequence of the lower level of the fluctuation background developed in the 3-D system with respect to that one developed in a two-dimensional system with equivalent characteristics.** The obliquely-propagating modes are strong enough and compete against the parallel-propagating modes in the linear stage. At later times ($t\Omega_p > 300$) the fluctuation energy decreases at larger propagation angles to the mean field.

Multi-channel coupling in the decay instability is identified in the spectral domain over the perpendicular wavenumbers. The energy spectrum for the magnetic field fluctuations, $\delta B^2(\omega, \boldsymbol{k})$, and that of the density, $\delta\rho^2(\omega, \boldsymbol{k})$, is plotted as slices in the frequency-wavenumber domain parallel to the mean magnetic field (Fig. 3 top), and in the wavevector domain spanning the perpendicular component $k_\perp$ and the parallel component $k_\parallel$ (Fig. 3 bottom). The spectral analysis is performed after the saturation of the instability at the latest time $t\Omega_p \approx 600$ of the simulation. The pump wave is in the Alfvén mode and propagates along the mean magnetic field ($k_\perp = 0$). Daughter Alfvén waves and daughter sound waves appear in the wavenumber-frequency spectrum at $k_\parallel V_A/\Omega_p \sim -0.2$ and $k_\parallel V_A/\Omega_p \sim 0.4$, respectively. The magnetic field and density fluctuations in Fig. 3(bottom

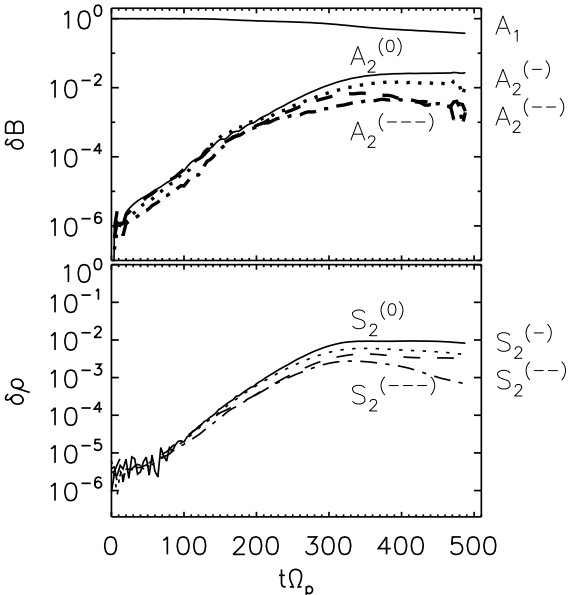

**Figure 2.** Time evolution of the magnetic field (top) and density (bottom) fluctuations: Alfvén field-aligned pump $A_1$, Alfvén daughter $A_2$ at selected perpendicular Fourier modes, and corresponding longitudinal daughter modes $S_2$ provided by the 3-D hybrid simulation. Symbols of '$A_2^{(0)}$', '$A_2^{(-)}$', '$A_2^{(--)}$','$A_2^{(---)}$', and '$S_2^{(0)}$', '$S_2^{(-)}$', '$S_2^{(--)}$','$S_2^{(---)}$' denote the mode m-numbers (-9;0,3,8,12) and (19;0,3,8,12), respectively. The propagation angle $\theta$ of the density daughter modes have values close to $10°$ (dotted), $20°$ (dashed), and $30°$ (dashed dotted).

panel) are analyzed at the frequency of the field-aligned daughter modes and differ in the perpendicular wavenumbers (or in the propagation angles). The obliquely-propagating Alfvén modes share the same frequency with that of the field-aligned Alfvén daughter wave. These fluctuations along the perpendicular direction to the mean magnetic field are daughter waves in the Alfvén mode resulted from the multi-channel coupling of the decay process. The daughter sound modes with oblique prop-

5  agation have the same frequency with the parallel-propagating daughter mode ($\omega \sim k_\parallel v_A \sqrt{\beta} \approx 0.05\Omega_p$) and thus fulfills the 3-wave coupling conservation law. One can mention that in the limit of low beta plasmas ($\beta \ll 1$), these oblique compressive waves are still normal modes of plasma.

## 2.2 Growth rate estimate

Growth rates for the individual decay instability can be computed from the hybrid simulation, and are compared to that of

10  the analytic dispersion analysis of two-fluid model developed by Viñas and Goldstein (1991a, b). **Briefly, the instability analysis of Viñas and Goldstein 1991ab consists in the following. By using the bi-fluid magnetohydrodynamic model, a linear perturbation of linear-mode waves is applied to study the stability of the large-amplitude circularly polarized Alfvén wave. Sideband electromagnetic waves are driven at upper and lower sideband frequencies obeying resonant wave-wave interaction rules. It is obtained a set of linear equations in the perturbed quantities satisfying the resonance**

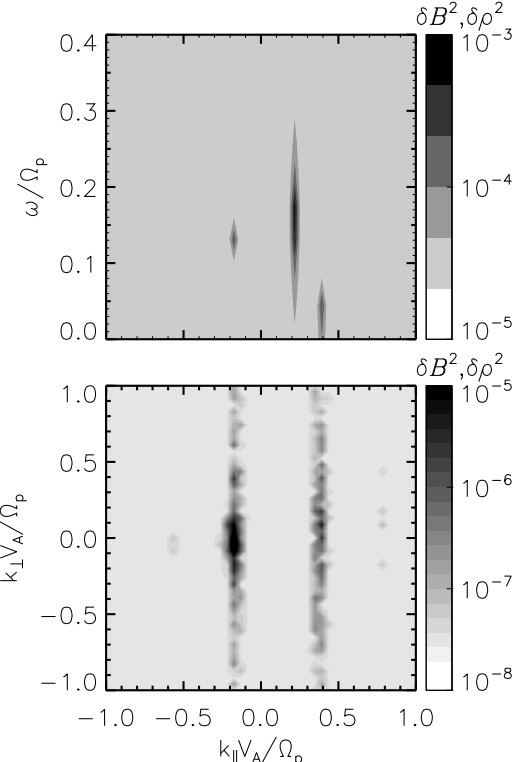

**Figure 3.** Spectrum for the magnetic field $\delta B^2(\omega, \boldsymbol{k})$ and density $\rho^2(\omega, \boldsymbol{k})$ fluctuations at the latest time of the simulation ($t\Omega_p \approx 600$). Slices at wavenumber $k_\perp$=0 (top panel) display the Alfvén pump wave $A_1$ ($k_\parallel V_A/\Omega_p \sim 0.2$), and the Alfvén $A_2^{(0)}$ ($k_\parallel V_A/\Omega_p \sim -0.2$) and the ion acoustic $S_2^{(0)}$ ($k_\parallel V_A/\Omega_p \sim 0.4$) daughter modes, in the frequency-wavenumber domain parallel to the mean magnetic field. Slices at the frequency of the Alfvén $A_2^{(0)}$ and density $S_2^{(0)}$ field-aligned modes are drawn in the perpendicular and parallel wavenumber domain in the bottom panel. The vertical extension on left ($k_\parallel V_A/\Omega_p \sim -0.2$) is for the Alfvén mode and that on right ($k_\parallel V_A/\Omega_p \sim 0.4$) is for the sound mode.

**condition while higher sidebands are neglected.** We consider the oblique modes at non-zero propagation angles to the mean magnetic field. The determinant of the matrix system is of order of 78 in terms of the frequency normalized by using the Alfvén speed $V_A$ and the ion gyro-frequency $\Omega_p$ for protons, $\hat{\omega} = \omega/(k_0 V_A) = \hat{\omega}_r + i\hat{\gamma}$. The elements of 6-by-6 order matrices can be found in the appendix in Viñas and Goldstein (1991a). The dispersion relation $\omega(\boldsymbol{k})$ and the growth rate $\gamma(\boldsymbol{k})$ are obtained by
5  solving the determinant of the matrix. We choose a low-beta value of 0.02 in the semi-analytic calculation. The solutions are numerically solved by using the symbolic calculator (Mathematica). We denote that the pump wave has the wavenumber $k_0$ and the frequency $\omega_0$, the longitudinal daughter wave $k$ and $\omega$, and the Alfvén (sideband) waves $\boldsymbol{k}^\pm$ and $\omega_\pm$. We assume that all the participating waves (pump wave, longitudinal daughter wave, and sideband wave) satisfying the coupling rules, that is,

$$\boldsymbol{k}^\pm = \boldsymbol{k}_0 \pm \boldsymbol{k} \tag{1}$$

10 $$\omega_\pm = \omega_0 \pm \omega \tag{2}$$

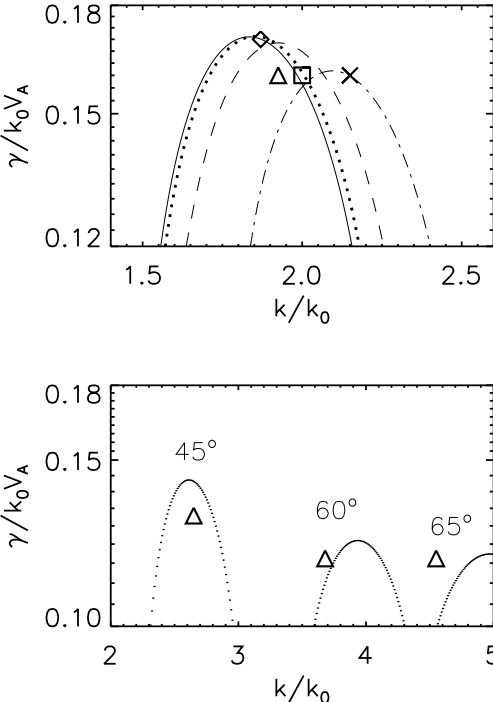

**Figure 4.** Growth rates of decay instability in the wavenumber domain of the daughter modes for propagation angles of $0°$ (solid), $10°$ (dotted), $20°$ (dashed), and $30°$ (dashed dotted) estimated by the **analytic dispersion** analysis (top panel). Overplotted by diamond, triangle, square, and cross symbols are the growth rates obtained in the 3-D hybrid simulation for the above propagation angles, respectively. Bottom panel shows the same comparison at larger propagation angles, $45°$, $60°$, and $65°$.

Figure 4 displays the growth rate $\hat{\gamma}$ as a function of the wavenumber for the parameters used in the hybrid simulation. Seven values of the propagation angle are chosen for the comparison: $\theta = \{0°, 10°, 20°, 30°, 45°, 60°, 65°\}$. The maximum growth rate falls at $k/k_0 \sim 1.9$ for the field-aligned decay ($\theta = 0°$). The solutions of the dispersion relation indicate that the growth rates slightly decreases at larger propagation angle. The result of the hybrid simulation is overall consistent with the **analytic**
5 **dispersion** analysis. The observed growth rates are overplotted in Fig. 4 and fit reasonably well the analytical prediction. At larger oblique angles, the prediction has larger deviations with respect to the simulation results. Kinetic effects are most probably responsible for such discrepancies between the MHD model and the hybrid simulation. The field-aligned decay weakly prevails, but the low and moderate oblique decay instabilities have growth rates close to the former one, confirming the increasing trend with beta decreasing (see e.g., Viñas and Goldstein, 1991b). Therefore, the multi-channel coupling decay
10 instability is regarded as a logical extension of the decay instability theory developed by Viñas and Goldstein (1991a, b) to simultaneous, independent, multiple decay channels.

## 3 Discussion and conclusions

**The damping of the oblique daughter waves observed in Fig. 2. can be explained in terms of the wave-particle processes (e.g., pitch-angle scattering, cyclotron resonance, or Landau damping) in competion with the plasma turbulence evolution. We have computed the time evolution of the spectrum of the magnetic field and density fluctuations. In the late stages of the fluctuation evolution, the one-dimensional reduced spectra (not shown) along the perpendicular wavenumber reveal turbulence cascades (by the fact that a power-law spectrum is being formed) starting from maxima at $k_\perp = 0$ specifically for low-beta plasma turbulence. The spectral slope along the parallel wavenumber is flatter in the low wavenumber range and becomes steeper at larger values beyond the pump wavenumber. This spectral anisotropy at low and moderate wavenumbers favors the parallel propagating modes which remain persistent at the latest stage of the decay process.**

The oblique electromagnetic daughter waves in Fig. 3 are followed by corresponding fluctuations in the wavenumber-frequency density spectrum (not shown). Viñas and Goldstein (1991a, b) in their analytic study found that the coupling between the electrostatic and electromagnetic solutions increases with the increase of the propagation angle of the daughter wave. On the basis of this remark, we naively interpret this result by interchanging the density daughter modes with the sideband electromagnetic waves. At large oblique propagation angles, the density daughter modes are electromagnetic waves with fluctuation both in density and electromagnetic fields while the sideband waves become electrostatic waves. Thus, the three-wave coupling rule is fulfilled. The obliquely-propagating modes develop into a turbulent cascade which saturates the instability at a stage when the Alfvén pump wave still dominates the spectrum. A modulation-like process is observed at early times close to the pump wavenumber and saturates before the saturation of the decay instability. The modulation is exclusively oblique and can be reminiscent of the beat instability. This additional parametric instability and the consequences on the decay instability and its early saturation are yet under study.

Matteini et al. (2010a) firstly noticed the occurence of a perpendicular spectrum of daughter waves developed at the decay of a linearly polarized Alfvén pump wave with oblique direction of propagation to the background magnetic field. The authors concluded that the generation of obliquely-propagating daughter waves is a consequence of the obliqueness of the Alfvén pump wave, i.e., its finite perpendicular wavenumber, in the framework of the 3-wave coupling process. Our 3-D treatment is simplified with respect to that former study in the sense that the pump wave is circularly polarized and propagates parallel to the mean magnetic field. Thus we can make useful the predictions of the MHD analytic analysis for the dispersion relations and the growth rates of the decay instability at different propagation angles. The dispersion relation for left-handed circular polarization, $(kV_A/\Omega_p)^2 = (\omega/\Omega_p)^2/(1 - \omega/\Omega_p)$, is satisfied for both the Alfvén pump wave (as imposed initial condition) and Alfvén daughter modes. The value of the pump wavenumber, $k_0 V_A/\Omega_p = 0.21$, is fairly high with respect to typical MHD scales but still it can be assumed in the dispersion-less range in the first order. The analytic dispersion analysis at weak and moderate oblique propagation angles provides the largest growth rates at approximative the same parallel wavenumber with that one predicted for the field-aligned decay. **This result can be easily understood by introducing the 3-wave coupling**

**equations and the dispersion relations,**

$$k_\parallel^- = k_0 - k_\parallel \tag{3}$$

$$k_\perp^- = -k_\perp \tag{4}$$

$$\omega^- = \omega_0 - \omega \tag{5}$$

**where $(k_0, \omega_0 = k_0 V_A)$, $(k_\parallel^-, k_\perp^-, \omega^- = -k_\parallel^- V_A)$, and $(k_\parallel, k_\perp, \omega)$ are the wavenumbers, frequencies, and dispersion relations characterizing the Alfvén pump, Alfvén daughter, and ion acoustic waves, respectively.** The MHD slow mode still preserves the field-aligned dispersion relation at small and moderate propagation angles in the limit of low beta plasmas, namely, $\omega_{slow} = k_\parallel V_A \sqrt{\beta}$ (here, $\beta = c_s^2 / V_A^2$). Therefore, the sound daughter modes with oblique propagation are normal modes of plasma like their counterpart oblique Alfvén modes and they fulfill similar dispersion relations, i.e, $\omega = k_\parallel c_s$. **Eqs. (3)-(5), the dispersion relations and some elementary algebra provide the parallel wavenumber of the oblique ion acoustic wave,**

$$k_\parallel = \frac{2k_0}{1 + c_s/V_A} \equiv k(0) \tag{6}$$

**where k(0) is wavenumber of the sound daughter driven by the field aligned decay (see e.g., Spangler et al., 1997).** The overall decay process is not controlled by the field-aligned decay but by the dispersion relation of the participating waves which drives the oblique decay to share identic parallel wavenumbers with those attained in the parallel decay. **The above rough evaluation is validated both by the analytic analysis in the MHD framework and by the hybrid simulation applied in the present study. In the analytic analysis, the evolution of the daughter waves is determined by the interaction with the field-aligned pump wave by constructing a system of quasi-linear equations which exhibits wave-wave couplings generating obliquely-propagating waves as the daughter component. The growth rate of the oblique modes has generally a maximum value at the parallel wavenumber prescribed by the maximum growth rate of the field-aligned decay instability and depends on the propagation angle.**

However, we still point out that the field-aligned decay remains in any case the fastest one, as shown in Fig. 4. Also, **this dynamics is different from the case of an oblique pump wave propagating at a given theta angle with respect to $B_0$, where the oblique mode's decay rate gamma is found to scale $\cos\theta_{kB}$, so controlled by the $k_\parallel$ projection of the initial oblique wavevector (Del Zanna, 2001). This suggests then that there are two possible ways of generating oblique modes from the parametric decay: "(1)" from a purely parallel mother wave, as in this study; "(2)" from an oblique pump wave (see e.g., Matteini et al., 2010a). In both cases the oblique modes grow at a rate that is smaller than the parallel decay. In our study the central role is played by the conservation of frequencies and the dispersion relations of the components involved in the 3-wave coupling. Matteini et al. (2010a) show that in their analysis, the central role in driving the transversal spectrum is played by the conservation of the momentum and the non-zero perpendicular projection of the pump wavevector. An obliquely-propagating (pump) wave is expected to be more compressive and it can generate a broad-band spectrum of compressive fluctuations. Thus, the configuration with an oblique pump**

wave should be more efficient in driving oblique modes, even though the daughter waves in our study are driven more early after a short time of nonlinear evolution. On the other hand, one should not ignore the major differences in the involved setups, e.g., the different polarization of the pump wave and different $\beta_e$ values (see discussion bellow). Besides the plasma beta, the wavenumber of the pump wave may play a significant role in the process of driving the transverse broadband modulation. In both studies, the medium is still weak dispersive ($k_0 V_A / \Omega_p \sim 0.2$). In a more dispersive medium such that one used by Verscharen et al. (2012), the dispersion relations probably cannot construct a transversal spectrum of waves for the decay of a field-aligned Alfvén wave.

Unlike MHD simulations, where the instability can saturate only through the steepening of the excited sound waves, in hybrid it saturates via particle trapping and phase-space modulation (see e.g., Matteini et al., 2010b). A consequence of this dynamics is a significant perturbation of the ion velocity distribution function, leading also to the generation of field-aligned beams (see e.g., Araneda et al., 2008). In low beta plasmas, the decay instability dominates over the other parametric instabilities. Plasma beta values for both ions and electrons play also an important role in the dynamics of the parametric decay. For instance, $\beta_e$ is responsible for activating different saturation mechanisms. Matteini et al. (2010b) reported that for cold fluid electrons ($\beta_e \sim 0$), the MHD saturation mechanism is recovered. At larger values (e.g., $\beta_e \sim 0.1$), the trapping and beam formation is in use. The electron plasma beta $\beta_e$ in our simulation has an intermediate value of 0.01. The parallel distribution in the phase space $z - v_\parallel$ (not shown) at a time still close to the linear phase of the instability ($t\Omega_p = 300$) suggests that particles are confined and accelerated in different regions along the parallel axis to the mean magnetic field. The phase-space modulation can be a signature for the instability saturation. There is no evidence of field-aligned velocity beam at the latest time of the simulation. The accelerated particles are thermalized via pitch-angle diffusion by the developed oblique modes.

A consequence of the oblique parametric decay in 3-D plasmas is **a more efficient heating of the ions**, see e.g., the pitch angle scattering study by Comişel et al. (2018). Their conclusion based on a hybrid simulation by using similar parameters with the actual run was that the plateau levels observed in the proton velocity distribution functions are driven by the pitch-angle diffusion of the ions by obliquely-propagating modes. The spectral analysis performed in Fig. 3 in the frequency-wavevector domain over a broad range of oblique propagation angles proves that the multi-channel decay of the parallel-propagating Alfvén pump wave is the source of obliquely-propagating daughter waves assumed to satisfy the resonance condition with ions. The former and the actual results reconfirm in particular the role of oblique compressive daughter waves in heating the protons parallel and perpendicular to the mean magnetic field according to the in-situ observation in solar wind plasmas, (see e.g., Marsch and Tu, 2001).

We have studied by 3-D hybrid simulations the decay of a large amplitude Alfvén pump wave expected to achieve the largest growth rate for parallel propagation and circularly polarization. Our conclusions are drawn below:

1. The parallel-propagating Alfvén wave decays into a transversal spectrum of daughter waves. The oblique decay is controlled by the growth rates of the decay instability which are significant larger at small and moderate propagation angles in low-beta plasmas, in agreement with the analytic analysis.

2. The transversal spectrum of daughter waves is controlled by the dispersion relation under the constraint of the 3-wave coupling process. Thus, the perpendicular alignment to the mean magnetic field originates in the dispersion state of the genuine circularly-polarized Alfvén wave.

Kinetic effects are assumed to become more important at larger propagation angles. Their role correlated with the value of plasma $\beta$ parameter will be subject for a forthcoming study of the decay instability in three-dimensional plasmas.

*Competing interests.* The authors declare that they have no conflict of interest.

*Acknowledgements.* HC acknowledges Yasuhiro Nariyuki for fruitful discussions, and the hospitality at the University of Toyama. This work is financially supported by a grant of the Deutsche Forschungsgemainschaft (DFG grant MO539/20-1). We acknowledge John von Neumann Institute for Computing (NIC) by providing computing time on the supercomputer JURECA at Juelich Supercomputer Centre (JSC).

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
