# Peer review of "Multi-channel coupling of decay instability in three-dimensional low-beta plasma"

_Annales Geophysicae, 2019_

## Referee Comment (RC1) · Anonymous Referee #1 · 20 Apr 2019

The authors of the manuscript "Multi-channel coupling of decay instability in three-dimensional low-beta plasma" discuss their numerical study of the parametric decay of circularly polarized Alfvén waves by means of hybrid plasma simulations. They investigate the role and evolution of oblique daughter waves generated by the decay instability and find an agreement between their results and analytical predictions. The manuscript is well written and describes the analysis in sufficient detail to be reproducible. Parametric decay is a candidate mechanism to explain the onset of plasma turbulence in the solar wind. This topic is of great interest to the solar, heliospheric, and geophysics communities. This work is thus highly suitable for publication in Annales Geophysicae. I identify a couple of minor shortcomings that require a revision of the submitted manuscript though. Until these comments have been addressed, I do not

recommend publication of this manuscript.

1) The Introduction section requires significantly more references both on beam instabilities and the parametric decay. As it stands, the manuscript does not provide the reader with sufficient context for the presented analysis.

2) On page 2, line 3: The authors refer to analytical treatments of parametric instabilities without giving references. It would be useful for the reader to have a couple of references that follow the analytical route described by the authors.

3) On page 4, line 17: The authors describe the decrease in amplitude of the oblique daughter waves. It would be interesting to discuss possible reasons for this behavior. Does this point relate to the development of a turbulent cascade that the authors describe in their Conclusions section?

4) On page 4, line 29: I recommend to write "$\omega\sim k_{\parallel} v_A \sqrt{beta}\approx 0.05 \Omega_p$" for the dispersion relation of the daughter wave at hand.

5) On pages 4 and 5, the authors refer to the model by Vinas and Goldstein as a "linear" analysis. This sounds a bit misleading since parametric decay is per se a nonlinear process. I recommend that the authors explain the "linear" nature of the Vinas & Goldstein analysis in some more detail.

6) On page 8, line 8, the authors give a dispersion relation for Alfvén waves. This expression clearly uses normalized definitions of k and omega. It would be good to give this normalization, or even better to re-write the equation using the normalization in terms of v_A and Omega_p. In a similar way, the dispersion relation given in line 13 requires an additional factor v_A on its right-hand side.

7) On page, line 21: The authors use the term "stochastic heating". I do not think the authors really refer to what is commonly described as "stochastic heating" at this point. It appears they refer to resonant wave-particle interactions and quasilinear diffusion

which are different from stochastic heating.
* * *

---

## Author Comment (AC1) · 16 May 2019

**Reply to referee comments**

Multi-channel coupling of decay instability in three-dimensional low-beta plasma

Manuscript ID: angeo-2018-14 (AngeoComm)
H. Comişel, Y. Narita, and U. Motschmann

We thank very much reviewer for reading our manuscript and raising helpful questions and suggestions. Here we give our reply. The changes in the manuscript are marked by bold letters.

*Reviewer:*
*The authors of the manuscript "Multi-channel coupling of decay instability in three-dimensional low-beta plasma" discuss their numerical study of the parametric decay of circularly polarized Alfvén waves by means of hybrid plasma simulations. They investigate the role and evolution of oblique daughter waves generated by the decay instability and find an agreement between their results and analytical predictions. The manuscript is well written and describes the analysis in sufficient detail to be reproducible. Parametric decay is a candidate mechanism to explain the onset of plasma turbulence in the solar wind. This topic is of great interest to the solar, heliospheric, and geophysics communities. This work is thus highly suitable for publication in Annales Geophysicae. I identify a couple of minor shortcomings that require a revision of the submitted manuscript though. Until these comments have been addressed, I do not recommend publication of this manuscript.*

*1) The Introduction section requires significantly more references both on beam instabilities and the parametric decay. As it stands, the manuscript does not provide the reader with sufficient context for the presented analysis.*

==================================================================

Reply:
We follow the reviewer's remark.

Changes in the manuscript:
"Narita, Y., Glassmeier, K.-H., Fränz, M., Nariyuki, Y., and Hada, T.: Observations of linear and nonlinear processes in the foreshock wave evolution, Nonlin. Processes Geophys., 14, 361-371, 2007.

Gary, S.P.: Electromagnetic ion/ion instabilities and their consequences in space plasmas: a review. Space Sci. Rev. 56, 373, 1991.

Akimoto, K., Winske, D., Gary, S.P., and Thomsen, M.F.: Nonlinear evolution of electromagnetic ion beam instabilities, J. Geophys. Res., 98, 1419-1433, doi:10.1029/92JA02345, 1993.

Gomberoff, K., Gomberoff, L. and, Astudillo, H.F.: Ion-beam-plasma electromagnetic instabilities, F. Plasma Physics, 64, 1, 75-87, 2000.

Wang, X.Y. and Lin, Y.: Generation of non-linear Alfvén and magnetosonic waves by beam-plasma interaction, Phys. Plasmas, 10, 9, doi:10.1063/1.1599359,2003.

Li, H., Pang, Y., Huang, S., Zhou, M., Deng, X., Yuan, Z., Wang, D., and Li, H.M.:The turbulence evolution in the high $\beta$ region of the Earth's foreshock, J. Geophys. Res., 118, 7151-7159, doi:10.1002/2013JA019424, 2013.

Terasawa, T., Hoshino, M., Sakai, J.-I., and Hada, T.: Decay instability of finite-amplitude circularly polarized Alfven Waves: A numerical simulation of stimulated Brillouin scattering, J. Geophys. Res., 91, 4171-4187,

https://doi.org/10.1029/JA091iA04p04171, 1986.

Hoshino, H. and Goldstein, M. L.: Time evolution from linear to nonlinear stages in magnetohydrodynamic parametric instabilities, Phys. Fluids B: Plasma Physics 1, 1405, https://doi.org/10.1063/1.858971, 1989.

5   Spangler, S. R., Leckband, J.A., and Cairns, I. H.: Observations of the parametric decay instability of nonlinear magnetohydrodynamic waves, Phys. Plasmas, 4, 846, https://doi.org/doi:10.1063/1.872183, 1997.

Nariyuki, Y., and Hada, T.: Remarks on nonlinear relation among phases and frequencies in modulational instabilities of
10  parallel propagating waves, Nonlin. Processes Geophys., 13, 425-441, 2006.

Bekhor, S. H. and Drake, R. P.: Plasma heating via parametric beating of Alfvén waves, with heliospheric applications, Phys. Plasmas 10, 4800, https://doi.org/10.1063/1.1619975, 2003.

15  Inhester, B.: A drift-kinetic treatment of the parametric decay of large-amplitude Alfvén waves, J. Geophys. Res., 95, 10, 525-10, 539, 1990.

Vasquez, B.J.: Simulation study of the role of ion kinetics in low frequency wave train evolution, J. Geophys. Res., 100, 1779-1792, 1995."

20  *Reviewer:*
*2) On page 2, line 3: The authors refer to analytical treatments of parametric instabilities without giving references. It would be useful for the reader to have a couple of references that follow the analytical route described by the authors.*

Reply:
25  Done.

Changes in the manuscript:
The following references are introduced:
"Derby, N. F. J.: Modulational instability of finite amplitude circularly polarized Alfvén waves, Astrophys. J., 224, 1013, 1978.

30  Goldstein, M. L.: An instability of finite-amplitude circularly polarized Alfvén waves, Astrophys. J., 219, 700, 1978.

Longtin, M., and Sonnerup, B. U. Ö.: Modulational instability of circularly polarized Alfvén waves, J. Geophys. Res., 91, 6816, 1986.

35  Wong, H. K., and Goldstein, M. L.: Parametric instabilities of circularly polarized Alfvén waves including dispersion, J. Geophys. Res., 91, 5617, 1986.

Hollweg, J. V.: The beat, modulational and decay instabilities of a circularly polarized Alfvén wave, J. Geophys. Res., 99,
40  23,431, 1994.

Ruderman, M. S., and Simpson, D.: The stability of parallel-propagating circularly polarized Alfvén waves revisited, J. Plasma Phys., 70, 143, doi:10.1017/S0022377803002599, 2004.

45  Araneda, J.A., Marsch, E., and Viñas, A.F.: Collisionless damping of parametrically unstable Alfvén waves, J. Geophys. Res., 112, A04104, doi:10.1029/2006JA011999, 2007.

Brodin, G., and Stenflo, L.: Three-wave coupling coefficients for perpendicular wave propagation in a magnetized plasma, Phys. Plasmas, 22, 104503, http://dx.doi.org/10.1063/1.4934938, 2015."

*Reviewer:*
*3) On page 4, line 17: The authors describe the decrease in amplitude of the oblique daughter waves. It would be interesting to discuss possible reasons for this behavior. Does this point relate to the development of a turbulent cascade that the authors describe in their Conclusions section?*

Reply:
The damping of the oblique daughter waves observed in Fig. 2. can be explained in terms of the wave-particle processes (e.g., pitch-angle scattering, cyclotron resonance, or Landau damping) in competion with the plasma turbulence evolution. We have computed the time evolution of the spectrum of the magnetic field and density fluctuations. In the late stages of the fluctuation evolution, the one-dimensional reduced spectra along the perpendicular wavenumber reveal turbulence cascades (by the fact that a power-law spectrum is being formed) starting from maxima at $k_\perp = 0$ specifically for low-beta plasma turbulence. The spectral slope along the parallel wavenumber is flatter in the low wavenumber range and becomes steeper at larger values beyond the pump wavenumber. This spectral anisotropy favors the parallel propagating modes which remain persistent at the latest stage of the decay process.

Changes in the manuscript:
Page 7, Line 10 to Page 8, Line 7:
The damping of the oblique daughter waves observed in Fig. 2. can be explained in terms of the wave-particle processes (e.g., pitch-angle scattering, cyclotron resonance, or Landau damping) in competion with the plasma turbulence evolution. We have computed the time evolution of the spectrum of the magnetic field and density fluctuations. In the late stages of the fluctuation evolution, the one-dimensional reduced spectra (not shown) along the perpendicular wavenumber reveal turbulence cascades (by the fact that a power-law spectrum is being formed) starting from maxima at $k_\perp = 0$ specifically for low-beta plasma turbulence. The spectral slope along the parallel wavenumber is flatter in the low wavenumber range and becomes steeper at larger values beyond the pump wavenumber. This spectral anisotropy at low and moderate wavenumbers favors the parallel propagating modes which remain persistent at the latest stage of the decay process.

*Reviewer:*
*4) On page 4, line 29: I recommend to write "$\omega \sim k_\parallel v_A \sqrt{beta} \approx 0.05\Omega_p$" for the dispersion relation of the daughter wave at hand.*

Reply:
Done.

Changes in the manuscript:
Page 5, Line 2,
$$|\omega| \sim \sqrt{\beta} \approx 0.05\Omega_p \rightarrow \omega \sim k_\parallel v_A \sqrt{\beta} \approx 0.05\Omega_p$$

*Reviewer:*
*5) On pages 4 and 5, the authors refer to the model by Vinas and Goldstein as a "linear" analysis. This sounds a bit misleading since parametric decay is per se a nonlinear process. I recommend that the authors explain the "linear" nature of the Vinas & Goldstein analysis in some more detail.*

Reply:
By linear analysis we refer the perturbative approach used to study the stability of the large-amplitude circularly polarized Alfvén wave conducting to a set of linear equations in the perturbed quantities. Meanwhile, higher sidebands are neglected. In order to avoid any confusion, we replace "linear analysis" by "instability analysis" or "analytic dispersion analysis". We introduce the following comment in the manuscript.

Changes in the manuscript:
Page 5, Line 6 to Line 11,
"Briefly, the instability analysis of Viñas and Goldstein 1991ab consists in the following. By using the bi-fluid magnetohydro-dynamic model, a linear perturbation of linear-mode waves is applied to study the stability of the large-amplitude circularly

polarized Alfvén wave. Sideband electromagnetic waves are driven at upper and lower sideband frequencies obeying resonant wave-wave interaction rules. It is obtained a set of linear equations in the perturbed quantities satisfying the resonance condition while higher sidebands are neglected."

Page 7, Caption of Fig. 4,

"linear analysis" $\rightarrow$ "analytic dispersion analysis"

Page 7, Line 1,

"linear analysis" $\rightarrow$ "analytic dispersion analysis"

*Reviewer:*

*6) On page 8, line 8, the authors give a dispersion relation for Alfvén waves. This expression clearly uses normalized definitions of k and omega. It would be good to give this normalization, or even better to re-write the equation using the normalization in terms of $v_A$ and $Omega_p$. In a similar way, the dispersion relation given in line 13 requires an additional factor $v_A$ on its right-hand side.*

Reply:

Done.

Changes in the manuscript:

Page 8, Line 26,

$k^2 = \omega^2/(1-\omega) \rightarrow (kV_A/\Omega_p)^2 = (\omega/\Omega_p)^2/(1-\omega/\Omega_p)$

Page 8, Line 31,

$\omega_{slow} = k_\parallel \sqrt{\beta} \rightarrow \omega_{slow} = k_\parallel V_A \sqrt{\beta}$

*Reviewer:*

*7) On page 8, line 21: The authors use the term "stochastic heating". I do not think the authors really refer to what is commonly described as "stochastic heating" at this point. It appears they refer to resonant wave-particle interactions and quasilinear diffusion which are different from stochastic heating.*

Reply:

We agree the referee's comment.

Changes in the manuscript:

Page 9, Line 4 is modified as follows,

[revised manuscript text omitted]

---

## Referee Comment (RC2) · Anonymous Referee #1 · 20 May 2019

The authors have addressed all of my comments and revised their manuscript appropriately. I'm now happy to recommend publication of this manuscript as an article in Annales Geophysicae.

---

## Referee Comment (RC3) · Anonymous Referee #2 · 19 Jul 2019

Dear Editor,

This is my review of the manuscript "Multi-channel coupling of decay instability in three-dimensional low-beta plasma" by Comisel, Narita and Motschmann.

The manuscript describes a study of the parametric decay instability of a mother (pump) Alfvén wave and the related wave-wave couplings in a fully 3-D hybrid simulation (kinetic ions and fluid electrons). The simulation analysis is overall quite concise, however it includes also a quite detailed comparison with linear theory predictions. The main result of the study is the generation of oblique modes from a purely parallel mother wave, whose growth rate are in reasonable good agreement with linear predictions and with some larger discrepancies at large angles.

[Figure]

I think the manuscript is potentially suitable for publication in Annales Geophysicae, however I think it would also benefit from some improvements in the presentation and discussion of the results, and some clarification in one of the figures.

Below I list my comments and suggestions in more details.

-

1. Authors conclude that: "The overall decay process is not controlled by the field-aligned decay but by the dispersion relation of the participating waves which drives the oblique decay to share identical parallel wavenumbers with those attained in the parallel decay" My understanding of their statement is that the growth rate of the oblique modes is not directly related to (i.e., a simple function of) the growth rate of the parallel mode. Is this interpretation correct? However, it should be still highlighted that the field aligned decay remains in any case the fastest one, as shown in Fig.4.

Also, this dynamics is different from the case of an oblique pump wave propagating at some theta angle with respect to $B\_0$, where the oblique mode's decay rate gamma is found to scale $\sim\cos(\text{theta})$, so controlled by the $k\_\parallel$ projection of the initial oblique $k$ (Del Zanna GRL 2001).

This suggests then that there are 2 possible ways of generating oblique modes from the parametric decay: 1) from a purely parallel mother wave, as in this study; 2) from an oblique pump wave (e.g. Matteini et al. GRL 2010). In both cases the oblique modes grow at a rate that is smaller than the parallel decay; it would be interesting then to establish which configuration would be more efficient in driving a transverse broadband modulation for, e.g., a given amplitude of the mother wave. Can authors discuss this possible competition in more detail?

2. About Figure 3. Bottom panels show the spectral pattern of both daughter waves (sound and Alfvén). Different peaks are labelled according to their 3-wave coupling described in Fig.1. However, there is no indication about the parallel $A\hat{\ }0$ and $S\hat{\ }0$

modes (present in the top panels). On the other hand, the most powerful signatures in the bottom panels are not labeled; should one conclude that they are the daughters labeled with "0" in Fig. 1? If so, why those modes have a non-zero k_perp in spectra of Fig. 4 (k_perp~0.03)? Does it mean that the parallel daughters of the top panels are not exactly field-aligned and have a finite k_perp? This could be an interesting result.

In any case, I think it would be useful if authors could extend the axis of the lower panels in order to include the k_perp=0 condition and show the reader the exact location of the "0" modes and if they lie on the k_|| axis as expected.

3. What is the exact amplitude of the initial wave? This can be approximately inferred from Fig.2, however the information should appear clearly in the text. Indeed, the daughter and sound waves saturate at a surprisingly low level (delta_b~delta_rho~0.01). Do the authors have an explanation for this? Does it depend on the - quite high - amplitude of the mother wave? or is it for a different reason (see next comment)?

4. This study is performed with an hybrid model, retaining ion kinetics, but through the manuscript no aspects about ion dynamics are mentioned or discussed. For example the fact that the instability saturates when the pump wave is still dominant could be a consequence of the kinetic nature of the plasma. Unlike MHD, where the instability can saturate only through the steepening of the excited sound waves, in hybrid it saturates via particle trapping and phase-space modulation (e.g. Matteini et al. JGR 2010). A consequence of this dynamics is a significant perturbation of the ion VDF, leading also to the generation of field-aligned beams (e.g. Araneda et al. 2008). I think it would be useful to add some information about the evolution of the particle VDF during the process, and possibly show it in a figure to also exploit the advantage of the hybrid model over fluid ones.

---

## Author Comment (AC3) · 5 Aug 2019

We thank the reviewer for the reply.
* * *

---

## Author Response (AR1)

**Reply to referee comments**

Multi-channel coupling of decay instability in three-dimensional low-beta plasma

Manuscript ID: angeo-2018-14 (AngeoComm)
H. Comişel, Y. Narita, and U. Motschmann

We thank very much the reviewer for reading our manuscript and raising helpful questions and valuable suggestions. Here we give our reply. The changes in the manuscript are marked in boldface.

*Reviewer:*
*1(a). Authors conclude that: "The overall decay process is not controlled by the field-aligned decay but by the dispersion relation of the participating waves which drives the oblique decay to share identical parallel wavenumbers with those attained in the parallel decay" My understanding of their statement is that the growth rate of the oblique modes is not directly related to (i.e., a simple function of) the growth rate of the parallel mode. Is this interpretation correct?*

Reply:
In order to answer the reviewer's question, we introduce the 3-wave coupling equations in more details,

$$k_\parallel^- = k_0 - k_\parallel \tag{1}$$

$$k_\perp^- = -k_\perp \tag{2}$$

$$\omega^- = \omega_0 - \omega \tag{3}$$

where $(k_0, \omega_0 = k_0 V_A)$, $(k_\parallel^-, k_\perp^-, \omega^- = -k_\parallel^- V_A)$, and $(k_\parallel, k_\perp, \omega)$ are the wavenumbers, frequencies, and dispersion relations characterizing the Alfvén pump, Alfvén daughter, and ion acoustic waves, respectively. Eqs. (1)-(3) and the assumption $\omega \sim k_\parallel c_s$ (available in low $\beta$ plasma) provide after some elementary algebra, the parallel wavenumber of the oblique ion acoustic wave,

$$k_\parallel = \frac{2k_0}{1 + c_s/V_A} \equiv k(0) \tag{4}$$

where k(0) is the wavenumber of the sound daughter driven by the field aligned decay, see e.g., Spangler et al., (1997).

The above rough evaluation is validated both by the analytic analysis in the MHD framework and by the hybrid simulation applied in the present study. In the analytic analysis, the evolution of the daughter waves is determined by the interaction with the field-aligned pump wave by constructing a system of quasi-linear equations which exhibits wave-wave couplings generating obliquely-propagating waves as the daughter component. The growth rate of the oblique modes has generally a maximum value at the parallel wavenumber prescribed by the maximum growth rate of the field-aligned decay instability and depends on the propagation angle.

Changes in the manuscript:
We rearrange the related comments in the former manuscript and introduce new comments as below,
Page 8, Line 34 to Page 9, Line 8:
" This result can be easily understood by introducing the 3-wave coupling equations and the dispersion relations,

$$k_\parallel^- = k_0 - k_\parallel$$

$$k_\perp^- = -k_\perp$$

$$\omega^- = \omega_0 - \omega$$

where $(k_0, \omega_0 = k_0 V_A)$, $(k_\parallel^-, k_\perp^-, \omega^- = -k_\parallel^- V_A)$, and $(k_\parallel, k_\perp, \omega)$ are the wavenumbers, frequencies, and dispersion relations characterizing the Alfvén pump, Alfvén daughter, and ion acoustic waves, respectively."

Page 9, Line 12 - Line 15:

"Eqs. (3)-(5), the dispersion relations and some elementary algebra provide the parallel wavenumber of the oblique ion acoustic wave,

$$k_\parallel = \frac{2k_0}{1 + c_s/V_A} \equiv k(0)$$

where k(0) is wavenumber of the sound daughter driven by the field aligned decay, see e.g., Spangler et al. (1997)."

Page 9, Line 17 - Line 23:

" The above rough evaluation is validated both by the analytic analysis in the MHD framework and by the hybrid simulation used in the present study. In the analytic analysis, the evolution of the daughter waves is determined by the interaction with the field-aligned pump wave by constructing a system of quasi-linear equations which exhibits wave-wave couplings generating obliquely-propagating waves as the daughter component. The growth rate of the oblique modes has generally a maximum value at the parallel wavenumber prescribed by the maximum growth rate of the field-aligned decay instability and depends on the propagation angle."

*Reviewer:*
*1(b). However, it should be still highlighted that the field aligned decay remains in any case the fastest one, as shown in Fig.4.*
*Also, this dynamics is different from the case of an oblique pump wave propagating at some theta angle with respect to $B_0$,*
*where the oblique mode's decay rate gamma is found to scale cos(theta), so controlled by the $k_\parallel$ projection of the initial oblique*
*k (Del Zanna GRL 2001). This suggests then that there are 2 possible ways of generating oblique modes from the parametric*
*decay: 1) from a purely parallel mother wave, as in this study; 2) from an oblique pump wave (e.g. Matteini et al. GRL 2010).*
*In both cases the oblique modes grow at a rate that is smaller than the parallel decay;*

Reply:
Agreed. We changed the section 3 accordingly.

Changes in the manuscript:
Page 9, Line 24 - Line 30:

"However, we still point out that the field-aligned decay remains in any case the fastest one, as shown in Fig. 4. Also, this dynamics is different from the case of an oblique pump wave propagating at a given theta angle with respect to $B_0$, where the oblique mode's decay rate gamma is found to scale $\cos\theta_{kB}$, so controlled by the $k_\parallel$ projection of the initial oblique wavevector (Del Zanna GRL 2001). This suggests then that there are two possible ways of generating oblique modes from the parametric decay: "(1)" from a purely parallel mother wave, as in this study; "(2)" from an oblique pump wave (e.g. Matteini et al. GRL 2010). In both cases the oblique modes grow at a rate that is smaller than the parallel decay."

*Reviewer:*
*1(c). ; it would be interesting then to establish which configuration would be more efficient in driving a transverse broadband*
*modulation for, e.g., a given amplitude of the mother wave. Can authors discuss this possible competition in more detail?*

Reply:
In our study the central role is played by the conservation of frequencies and the dispersion relations of the components involved in the 3-wave coupling. Matteini et al. GRL (2010a) show that in their analysis, the central role in driving the transversal spectrum is played by the conservation of the momentum and the non-zero perpendicular projection of the pump wavevector. An obliquely-propagating (pump) wave is expected to be more compressive and it can generate a broad-band spectrum of

compressive fluctuations. Thus, the configuration with an oblique pump wave should be more efficient in driving oblique modes, even though the daughter waves in our study are driven more early after a short time of nonlinear evolution. On the other hand, one should not ignore the major differences in the involved setups, e.g., the different polarization of the pump wave and different $\beta_e$ values (see discussion bellow). Besides the plasma beta, the wavenumber of the pump wave may play a significant role in the process of driving the transverse broadband modulation. In both studies, the medium is still weak dispersive ($k_0 V_A / \Omega_p \sim 0.2$). In a more dispersive medium such that one used by Verscharen et al. (2012), the dispersion relations probably cannot construct a transversal spectrum of waves for the decay of a field-aligned Alfvén wave.

Changes in the manuscript:
Page 9, Line 30 to Page 10, Line 7:
"In our study the central role is played by the conservation of frequencies and the dispersion relations of the components involved in the 3-wave coupling. Matteini et al. GRL 2010a show that in their analysis, the central role in driving the transversal spectrum is played by the conservation of the momentum and the non-zero perpendicular projection of the pump wavevector. An obliquely-propagating (pump) wave is expected to be more compressive and it can generate a broad-band spectrum of compressive fluctuations. Thus, the configuration with an oblique pump wave should be more efficient in driving oblique modes, even though the daughter waves in our study are driven more early after a short time of nonlinear evolution. On the other hand, one should not ignore the major differences in the involved setups, e.g., the different polarization of the pump wave and different $\beta_e$ values (see discussion bellow). Besides the plasma beta, the wavenumber of the pump wave may play a significant role in the process of driving the transverse broadband modulation. In both studies, the medium is still weak dispersive ($k_0 V_A / \Omega_p \sim 0.2$). In a more dispersive medium such that one used by Verscharen et al. (2012), the dispersion relations probably cannot construct a transversal spectrum of waves for the decay of a field-aligned Alfvén wave."

*Reviewer:*
*2. About Figure 3. Bottom panels show the spectral pattern of both daughter waves (sound and Alfvén). Different peaks are labelled according to their 3-wave coupling described in Fig.1. However, there is no indication about the parallel $A^0$ and $S^0$ modes (present in the top panels). On the other hand, the most powerful signatures in the bottom panels are not labeled; should one conclude that they are the daughters labeled with "0" in Fig. 1? If so, why those modes have a non-zero $k_\perp$ in spectra of Fig. 4 ($k_\perp \sim 0.03$)? Does it mean that the parallel daughters of the top panels are not exactly field-aligned and have a finite $k_\perp$? This could be an interesting result. In any case, I think it would be useful if authors could extend the axis of the lower panels in order to include the $k_\perp = 0$ condition and show the reader the exact location of the "0" modes and if they lie on the $k_{||}$ axis as expected.*

Reply:
No, the most powerful signatures in the bottom panels of Fig. 3 are not the daughters labeled with "0" in Fig. 1. By labeling several modes in the bottom panels we suggested several possible 3-wave coupling processes while the mentioned modes cannot fulfill the same vector triangle with the pump wave. The daughter modes labeled with "0" are exact parallel-propagating waves ($k_\perp$=0). We did not show them in the bottom panels because the vertical axes have a logarithmic scale in order to delimitate the labels $A_2^{(...)}$ and $S_2^{(...)}$.

We follow the suggestion of the reviewer to include $k_\perp = 0$ and extend the axes for both positive and negative values of the perpendicular wavenumbers. We also flip the sign of frequency to comply the conventional wave-wave coupling diagram (parallelogram diagram) in the k-omega domain shown in Fig. 1. In this way the left-handed polarization of the involved waves is conventionally represented for positive values of the frequency.

Changes in the manuscript:
Figure 3 is updated.

*Reviewer:*
*3. What is the exact amplitude of the initial wave? This can be approximately inferred from Fig.2, however the information*

*should appear clearly in the text. Indeed, the daughter and sound waves saturate at a surprisingly low level ($\delta b \sim \delta \rho \sim 0.01$).*
*Do the authors have an explanation for this? Does it depend on the - quite high - amplitude of the mother wave? or is it for a*
*different reason (see next comment)?*

Reply:
The magnetic-field amplitude of the pump wave (normalized to the value of the background magnetic field) has a value of 0.2.
Thank you very much for pointing out this lacking information in the original manuscript.

We note that the overall background level of the 1-D reduced density spectrum of density fluctuation $\delta\rho(k_\parallel)$ is significantly
lower than the 1-D reduced density spectrum obtained from an equivalent 2-D simulation. However, the amplitude of the driven
daughter mode related to the background level is approximatively the same between the 3-D and 2-D setups. The same kind of
result is obtained for the 1-D reduced spectrum of the magnetic field fluctuations.

No, we do not think that the large amplitude of the Alfvén pump wave could have a significant role here. The analytic analyses,
see e.g., Viñas and Goldstein (1991b), show that the growth rate of the decay instability increases as the amplitude of the pump
wave is getting larger values, and a weaker slope is observed at about $\eta = \delta B / B_0 > 0.4$ for the parallel propagation case.
Naively speaking, one may expect that the amplitude of the daughter modes should be larger as well, as long as there is no
saturation of the instability with respect to the pump wave amplitude.

Changes in the manuscript:
Page 4, Line 9 - Line 10:
"The magnetic-field amplitude of the Alfvén pump wave (normalized to the value of the background magnetic field) has a
value of 0.2."
Page 4, Line 23 - Line 25:
"The low level of the amplitude of the magnetic field and density daughter modes in Fig. 2 could be a consequence of the
lower level of the fluctuation background developed in the 3-D system with respect to that one developed in a two-dimensional
system with equivalent characteristics."

*Reviewer:*
*4. This study is performed with an hybrid model, retaining ion kinetics, but through the manuscript no aspects about ion*
*dynamics are mentioned or discussed. For example the fact that the instability saturates when the pump wave is still dominant*
*could be a consequence of the kinetic nature of the plasma. Unlike MHD, where the instability can saturate only through the*
*steepening of the excited sound waves, in hybrid it saturates via particle trapping and phase-space modulation (e.g. Matteini*
*et al. JGR 2010). A consequence of this dynamics is a significant perturbation of the ion VDF, leading also to the generation*
*of field-aligned beams (e.g. Araneda et al. 2008). I think it would be useful to add some information about the evolution of the*
*particle VDF during the process, and possibly show it in a figure to also exploit the advantage of the hybrid model over fluid*
*ones.*

Reply:
We thank very much the reviewer for the suggestion.
In low beta plasmas, the decay instability dominates over the other parametric instabilities. Plasma beta values for both ions
and electrons play also an important role in the dynamics of the parametric decay. For instance, $\beta_e$ is responsible for activating
different saturation mechanisms. Matteini et al. (2010) JGR reported that for cold fluid electrons ($\beta_e \sim 0$), the MHD saturation
mechanism is recovered. At larger values (e.g., $\beta_e \sim 0.1$), the trapping and beam formation is in use.
The electron plasma beta $\beta_e$ in our simulation has an intermediate value of 0.01. Information on the particles has not been
saved from this simulation but particle data are available from a former 3-D simulation performed at a lower resolution at
time $t\Omega_p = 300$ and at the final time $t\Omega_p = 600$ (Comişel et al. 2018). The parallel distribution in the phase space $z - v_\parallel$ (not
shown) at a time still close to the linear phase of the instability ($t\Omega_p = 300$) suggests that particles are confined and accelerated
in different regions along the parallel axis to the mean magnetic field. The phase-space modulation can be a signature for the
instability saturation. There is no evidence of field-aligned velocity beam at the latest time of the simulation. The accelerated
particles are thermalized via pitch-angle diffusion by the developed oblique modes.

Changes in the manuscript:

Page 10, Line 8 - Line 20:

"Unlike MHD simulations, where the instability can saturate only through the steepening of the excited sound waves, in hybrid it saturates via particle trapping and phase-space modulation (e.g. Matteini et al. JGR 2010). A consequence of this dynamics is a significant perturbation of the ion velocity distribution function, leading also to the generation of field-aligned beams (e.g. Araneda et al. 2008). In low beta plasmas, the decay instability dominates over the other parametric instabilities. Plasma beta values for both ions and electrons play also an important rule in the dynamics of the parametric decay. For instance, $\beta_e$ is responsible for activating different saturation mechanisms. Matteini et al. (2010) JGR reported that for cold fluid electrons ($\beta_e \sim 0$), the MHD saturation mechanism is recovered. At larger values (e.g., $\beta_e \sim 0.1$), the trapping and beam formation is in use. The electron plasma $\beta_e$ in our simulation has an intermediate value of 0.01. The parallel distribution in the phase space $z - v_\parallel$ (not shown) at a time still close to the linear phase of the instability ($t\Omega_p = 300$) suggests that particles are confined and accelerated in different regions along the parallel axis to the mean magnetic field. The phase-space modulation can be a signature for the instability saturation. There is no evidence of field-aligned velocity beam at the latest time of the simulation. The accelerated particles are thermalized via pitch-angle diffusion by the developed oblique modes."

===================================================================